# Safety and Efficacy of Sodium and Potassium Arachidonic Acid Salts in the Young Pig

**DOI:** 10.3390/nu13051482

**Published:** 2021-04-27

**Authors:** Kaylee Hahn, Joseph R. Hardimon, Doug Caskey, Douglas A. Jost, Patrick J. Roady, J. Thomas Brenna, Ryan N. Dilger

**Affiliations:** 1Division of Nutritional Sciences, University of Illinois, Urbana, IL 61801, USA; khahn0114@gmail.com; 2Jost Chemical Co, St., Louis, MO 63114, USA; joe.hardimon@jostchemical.com (J.R.H.); doug.caskey@jostchemical.com (D.C.); doug.jost@jostchemical.com (D.A.J.); 3Veterinary Diagnostic Laboratory, College of Veterinary Medicine, University of Illinois, Urbana, IL 61802, USA; roady@illinois.edu; 4Department of Veterinary Clinical Medicine, College of Veterinary Medicine, University of Illinois, Urbana, IL 61802, USA; 5Dell Pediatric Research Institute, Department of Pediatrics, of Chemistry, and of Nutrition, University of Texas at Austin, Austin, TX 78723, USA; tbrenna@gmail.com; 6Division of Nutritional Sciences, Cornell University, Ithaca, NY 14853, USA; 7Department of Animal Sciences, University of Illinois, Urbana, IL 61801, USA

**Keywords:** arachidonic acid, docosahexaenoic acid, bioequivalence, infant nutrition, pig

## Abstract

Arachidonic acid (ARA; 20:4n6) and docosahexaenoic acid (DHA; 22:6n3) are polyunsaturated fatty acids (FA) naturally present in breast milk and added to most North American infant formulas (IF). We investigated the safety and efficacy of novel sodium and potassium salts of arachidonic acid as bioequivalent to support tissue levels of ARA comparable to the parent oil; *M. alpina* oil (Na-ARA and K-ARA) and including a Na-DHA group. Pigs of both sexes were randomized to one of five dietary treatments (*n* = 16 per treatment; 8 male and 8 female) from postnatal day 2 to 23. ARA and DHA were included as either triglyceride (TG) or salt. Target dietary ARA/DHA concentrations as percent of total FA by weight were as follows: TT (0.47 TG/0.32 TG), NaT (0.47 Na-salt/0.32 TG), KT (0.47 K-salt/0.32 TG), and Na0 (0.47 Na-salt/0.00), NaNa (0.47 Na-salt/0.32 Na-salt). The primary outcome in this study was bioequivalence of ARA brain accretion. Growth performance; blood and tissue fatty acid levels; liver histology; complete blood cell counts; and serum chemistries were all evaluated. Overall, diets containing test sources of ARA and DHA did not affect growth performance; liver histology; or substantially influence hematological outcomes as compared with TT. The results confirm that the use of Na and K salt forms of ARA yield bioequivalent ARA accretion in the cerebral cortex and retinal tissue compared to TG-ARA. These findings confirm that use of Na-ARA and K-ARA salts in the young pig was safe and nutritionally bioequivalent to TG-ARA for critical neural tissues.

## 1. Introduction

Arachidonic acid (ARA, 20:4n6) and docosahexaenoic acid (DHA, 22:6n3) are polyunsaturated fatty acids (PUFA) naturally present in breast milk and frequently supplemented in infant formulas (IF). Based on worldwide averages, concentrations of breast milk ARA and DHA are approximately 0.47 ± 0.13% and 0.32 ± 0.22%, as % of total fatty acids (FA) by weight, respectively [1]. Inclusion of ARA and DHA in IF typically reflects the concentrations observed in breast milk. While only present in low levels of the diet, both rapidly accumulate in the central nervous system during perinatal development, representing the two most abundant PUFA in the retina and brain [2,3]. In addition to tissue accretion, ARA and DHA function within the immune system, serving as precursors for predominantly pro-and anti-inflammatory compounds, respectively [4,5]. Infants are able to endogenously synthesize ARA and DHA from their essential fatty acid precursors, but the ability to do so appears limited [6]. In formula-fed infants, supplementation with preformed ARA and DHA is shown to yield more comparable cognitive, visual, and immune outcomes to that of a breastfed infant [7,8,9,10,11,12]. Preclinical animal models have also helped to demonstrate improved tissue accretion with the inclusion of preformed ARA and DHA [13,14,15]. Additionally, polymorphisms in the fatty acid desaturase (FADS) genes modify the synthesis of longer, more unsaturated PUFA from essential FA precursors. As such, infants with specific genotypes may require different amounts of ARA and DHA to meet physiologic needs [16]. Accordingly, both ARA and DHA are often considered conditionally essential during early life. 

FA are primarily consumed as triglycerides (TG), phospholipids (PL), or free fatty acids (FFA). Breast milk and IF provide nearly half of their energy as lipids, with more than 98% in the form of TG [17,18]. Gastric and lingual lipases are particularly important in accommodating this lipid load, as newborns produce lower levels of pancreatic lipase than adults [19,20]. Gastric lipase, however, appears to be less active against lipids contained in IF than breast milk [20]. Unlike IF, breast milk contains bile salt-stimulated lipase, which functions in the small intestine and helps compensate for limited pancreatic lipase production [21,22]. Most of the ARA and DHA in breast milk are found within the TG fraction and both make up a large proportion of the PL fraction, but the total concentration of PL in breast milk is much lower [23]. The form in which ARA and DHA are consumed, PL or TG, is shown to influence uptake and accretion in tissues, with animal studies demonstrating that those in the PL form are preferentially taken up by tissues compared with those in TG form [24,25,26]. To optimize ARA and DHA content in diet formulations, differences in bioavailability and tissue-specific accretion between lipid forms must be considered.

The addition of preformed ARA and DHA to IF in the United States began in 2001, and a number of products have received the designation of ‘generally recognized as safe’ (GRAS) for use in IF [27,28,29,30,31]. Most provide ARA derived from fungus (*Mortierella alpina*), and DHA from algal sources (*Schizochytrium* sp. and *Crypthecodinium cohnii*); PL forms extracted from egg yolk, and DHA from tuna oil are also available. There are no FFA forms of either ARA or DHA currently approved for use in IF. Little is known about the efficacy and use of ARA as FFA compared with alternative forms, but studies in adults show similar or improved bioavailability with DHA and eicosapentaenoic acid (EPA) when provided as FFA compared to TG or ethyl ester forms [32,33,34,35]. Given the similar metabolic pathways between these PUFA, ARA as an FFA may show similar differences.

The present study sought to evaluate the use of novel sodium and potassium salts of *M. alpina* oil (Na-ARA and K-ARA, respectively) for potential application as FA supplements for IF. These salts have the processing advantage of being solid flowing powders rather than liquids. Nutritionally, ARA salts dissociate into Na^+^/K^+^ and FFA in water, providing the benefit of not requiring digestive hydrolysis. This is of particular interest for IF design due to infants having lower levels of pancreatic lipase and gastric lipase that is less active against IF lipid, as previously discussed. The young pig was used as a preclinical model to evaluate the safety and efficacy of these Na-ARA and K-ARA salts in the presence of DHA, and demonstrate nutritional bioequivalence to ARA as TG to support tissue ARA accretion. The young pig is an ideal preclinical model for infant nutrition and early development studies due to similarities in anatomy, physiology, immunology, and metabolism [36,37,38,39,40,41,42]. Safety was assessed based on growth performance, tolerance, liver histology, and hematological panels. Bioequivalence was primarily assessed by ARA accretion in cerebral cortex and retinal tissues as compared with the ARA as TG. Additionally, FA concentrations were quantified in the liver, heart, skeletal muscle, plasma, and red blood cells (RBC) as secondary outcomes. We hypothesized that ARA salts would be nutritionally bioequivalent to ARA in the TG form, yielding similar FA profiles when provided alongside DHA, and we did not anticipate any safety issues due to ARA salt inclusion.

## 2. Materials and Methods

### 2.1. Animal Care and Housing

Eighty naturally-farrowed intact male and female pigs were obtained from a commercial swine farm (PIC 1050 genetics; Carthage Veterinary Service, Ltd., Carthage, IL, USA) and transported to the Piglet Nutrition and Cognition Laboratory on postnatal day (PND) 2. Pigs received a prophylactic antibiotic injection on PND 1 (EXCEDE^®^, Zoetis, Parsippany, NJ, USA). Upon entering the facility, animals were administered 5 mL of prophylactic Clostridium perfringens types C and D antitoxin subcutaneously and orally to avoid enterotoxemia (Colorado Serum Company, Denver, CO, USA). The study was completed in four cohorts (i.e., groups of pigs of similar background that were enrolled on study at the same time), with 16 pigs per treatment and an equal number of intact male and female pigs assigned within each treatment group. Pigs were selected across eight litters and distributed equally across all five dietary treatment groups. An oral electrolyte solution was provided if pigs experienced diarrhea for greater than three days (Bounce Back; Manna Pro Products, Chesterfield, MO, USA). Per veterinary recommendation, all pigs enrolled in cohort 3 were treated with an antibiotic injection on PND 10 (0.1 mL/kg body weight; EXCEDE^®^, Zoetis, Parsippany, NJ, USA). Animals were housed as previously described [43]. Pigs were humanely euthanized via CO_2_ asphyxiation on PND 23. All animal care and experimental procedures were in accordance with the National Research Council Guide for the Care and Use of Laboratory Animals and approved by the University of Illinois at Urbana-Champaign Institutional Animal Care and Use Committee (IACUC Protocol #17286).

### 2.2. Dietary Treatments

Custom milk replacer (MR) products (TestDiet; St. Louis, MO, USA) were designed based on the nutrient requirements for young pigs [44]. Soy protein isolate was used as the MR base to ensure no inherent ARA or DHA contribution; internal FA analyses of commercial pig MR powders revealed moderate levels of ARA (~0.1–0.3% total FA; data not shown). Powdered fat products were selected after internal analyses confirmed the absence of ARA and DHA (data not shown). Powdered TG-ARA and DHA (CABIO Bioengineering Co., Ltd., Wuhan, China) and experimental Na- and K-ARA-enriched free fatty acid products were provided by Jost Chemical Co (St. Louis, MO, USA). Samples of MR diets were collected shortly after initiating the live-animal study and stored in airtight glass containers away from light and at −20 °C until analysis.

Pigs were randomly allotted (*n* = 8 intact males and 8 females per treatment) to one of five isocaloric experimental milk replacers (Table 1) by initial body weight and litter [45]. The inclusion of ARA and DHA was in the form of either TG, the most frequently supplemented forms in IF, or as Na^+^/K^+^-FA salt form. Treatments targeted the following ARA/DHA concentrations as percent of total fatty acids by weight: TT (0.47 TG/0.32 TG), NaT (0.47 Na-salt/0.32 TG), KT (0.47 K-salt/0.32 TG), and Na0 (0.47 Na-salt/0.00), NaNa (0.47 Na-salt/0.32 Na-salt).

Water and reconstituted MR were supplied to the pigs ad libitum. Diets were reconstituted daily with 200 g of MR powder per 800 g water and distributed using an automated liquid feeding system over a 20-h period; the remaining 4 h were used to clean the feeding system and introduce fresh MR.

### 2.3. Growth and Wellness

Individual pigs and MR reservoirs were weighed daily to calculate average daily body weight gain (ADG) and average daily milk intake (ADMI; i.e., net disappearance of MR), respectively. Feed efficiency, gain to feed (G:F), was calculated as the ratio of grams of body weight gained to grams of MR consumed. Animals were observed two times a day for general health and wellness. Stool consistency was visually assessed and scored daily using the following scale: 1 = solid; 2 = semi-solid; 3 = loose; 4 = watery.

### 2.4. Hematological Panels

On PND 23, serum and whole blood were collected and submitted to the University of Illinois College of Veterinary Medicine Clinical Pathology Laboratory for serum chemistry profiles and complete blood cell counts with differentials. Whole blood was also collected for plasma and RBC membrane FA analysis. Whole blood was collected into 4-mL evacuated K_2_EDTA blood tubes (Becton Dickinson & Company, Franklin Lakes, NJ, USA, Cat No: 367835) using 21 gauge × 3.18 cm collection needles (Becton Dickinson & Company, Franklin Lakes, NJ, USA, Cat No: 368607). Samples were stored on ice then centrifuged to separate plasma and RBC (4 °C, 1250× *g*, 15 min; Allegra 6R centrifuge, Beckman Coulter Life Sciences, Indianapolis, IN, USA). Aliquots of plasma and RBC were stored at −80 °C. Serum samples were collected into 4-mL evacuated serum tubes (BD, Franklin Lakes, NJ, USA, Cat No: 367812) and left to clot for at least 30 min at room temperature. All samples were processed within 6 h of collection, and aliquots were stored at −80 °C.

### 2.5. Tissue Collection

Organ weights and tissue aliquots were collected on PND 23. Tissues included whole brain, liver, longissimus dorsi, heart, kidney, spleen, lung, and retina. Tissue samples were dissected using a scalpel and homogenous aliquots were snap-frozen in liquid nitrogen and stored at −80 °C.

### 2.6. Fatty Acid Analyses

Fatty acid composition of MR powder, as well as animal tissues and blood, were quantified by gas chromatography. Samples (~100 mg or 100 µL) were subjected to a one-step, direct transesterification procedure to generate fatty acid methyl esters (FAME) [46]. Tissue samples were lyophilized prior to transesterification as a precaution to prevent reaction interference from water. Two internal FA standards (11:0 and 23:0) were used within the assay, and a quantitative external standard was used to measure response factors for the flame ionization detector. FA of interest were identified and quantified by comparing retention times with known FA contained in a mixed standard (Supelco 37 Component FAME Mix; Supelco, Bellefonte, PA, USA). A sample peak was defined as being 3-times the height of background noise. Peak less than 3-times the background noise were considered not detectable.

Quantification of FAME was performed on gas chromatographers (either model 6890, Agilent, Santa Clara, CA or model 5890 Series II, Hewlett-Packard, Palo Alto, CA, USA) equipped with a 100-m × 0.25 mm i.d. × 0.2 µm film thickness capillary column (model SP-2560; Supelco, Bellefonte, PA, USA). Injection (25:1 split ratio) was performed at 240 °C and detection was performed at 245 °C. Starting column head pressure was 48 psi with a constant helium flow of 1.6 mL/min. The oven temperature program was as follows: 120 °C isothermal for 7 min, increased to 180 °C at 1 °C/min and held for 8 min, increased to 240 °C at 1 °C/min and held for 12 min, decreased to 120 °C at 20 °C/min and held for 9 min.

### 2.7. Diet Analysis

Diet samples were collected within a week of the first cohort of pigs being enrolled on study and stored at −20 °C in airtight containers away from light. Samples were cryogenically pulverized (6700 Freezer/Mill^®^, SPEX SamplePrep, Metuchen, NJ, USA) prior to FA analyses to ensure homogeneity and particle size needed to produce accurate analytical results. Additional information on dietary treatments and analysis is available in Appendix A.

### 2.8. Liver Histology

Livers were evaluated by a board-certified veterinary pathologist at the University of Illinois Veterinary Diagnostic Laboratory. Evaluations were made using 3 µm hematoxylin and eosin-stained sections prepared from formalized samples of the right lateral, left medial, and left lateral lobes of the liver (formalin solution, neutral buffered, 10%, Sigma-Aldrich, HT501128). Tissues were embedded using Leica Formula R paraffin (Leica Biosystems, Buffalo Grove, IL, USA). Tissues were placed down into a mold filled with molten paraffin, cassettes placed on the top of the mold, filled with more molten paraffin, then placed on a cold plate to solidify. Blocks were taken out from the mold and excess paraffin removed. Blocks were faced using a microtome to obtain a full face section. Ribbons were cut at 3 µm and placed on a warm water bath (45–50 °C). Sections free of folds and wrinkles were picked up onto slides, then placed into a slide dryer (60–70 °C) for 20–30 min. Slides were then stained with hematoxylin and eosin and coverslipped using an automated stainer and coverslipper, respectively (Sakura Finetek, Torrance, CA, USA).

### 2.9. Statistical Analysis

Power analysis. The number of piglets in each group was powered based on SD of cortex ARA in a similar study [47]. Tissue ARA is normally distributed with standard deviation 0.08% *w*/*w* or less. To detect a difference of 10%, e.g., 10 vs. 11% *w*/*w*, between experimental and control oils, 8 experimental piglets will be required for 81% power. The Type I error probability associated with this test of this null hypothesis is 0.05. Calculations were made using nQuery Advisor 3.0.

For the primary outcome of ARA accretion in the cerebral cortex, values from at least 11 pigs per treatment were included. Results demonstrate that this number of pigs was adequate to detect statistical differences in cerebral cortex ARA between dietary treatment groups. As described in more detail in below, those not provided dietary DHA (Na0) had elevated levels of cerebral cortex ARA compared to those fed NaT and KT.

All outcomes were analyzed by 1-way ANOVA using the MIXED procedure of SAS 9.4 (SAS Institute, Inc., Cary, NC, USA). Diet was included as a fixed effect; cohort and sex were included as random effects. Outliers were defined as values with a Studentized Residual with an absolute value of ≥3 and were removed prior to final analysis. Significance was accepted at *p* ≤ 0.05 and data are presented as least-squares means (LSM) with pooled standard errors of the mean (SEM).

### 2.10. Bioequivalence Assessment

Bioequivalence was assessed by generating 90% confidence intervals (CI) around the geometric least square mean ratio of tissue ARA percentage in the test diets (NaT, KT, Na0, NaNa) compared to the control diet (TT), consistent with similar studies evaluating alternate sources of ARA and DHA [47,48]. Confidence intervals were calculated from log-transformed ratios. Upper and lower bounds were back transformed by exponentiation and presented as percentages. Per FDA guidelines, bioequivalence was achieved if the 90% CI falls within 80–125% [49,50,51].

## 3. Results

### 3.1. Bioequivalence

The primary outcome of this trial was the bioequivalence of Na-ARA and K-ARA to TG-ARA for ARA brain accretion. All 90% CI for critical tissues, i.e., cerebral cortex and retina, fell within the FDA defined 80–125% limits for bioequivalence (Figure 1). Bioequivalence for secondary tissues was also evaluated, which exhibited wider CI ranges exceeding the 80–125% threshold (Figure 1).

### 3.2. Growth and Tolerance

Growth performance results are summarized in Table 2. There were no differences (*p* > 0.05) between treatment groups for body weight, ADG, ADMI, or G:F. Stool consistency was both normal (i.e., based on historical internal data for artificially-reared pigs) and comparable between treatment groups throughout the duration of the study (data not shown). While no differences were observed between treatments for raw organ weights (data not shown), a few differences were observed when compared as relative organ weights, specifically for the spleen and kidney. Pigs consuming diets Na0 and NaNa had similar relative spleen weights, but both were lower (*p* < 0.05) than those fed TT, NaT, and KT, which had similar relative kidney weights. Pigs fed NaT and NaNa had similar relative kidney weights, but both were higher (*p*
*<* 0.05) than those fed TT, KT, and Na0, which had similar relative kidney weights.

### 3.3. Hematological Panels

Few differences were observed in hematological outcomes between treatment groups. Differences were observed for serum phosphorus (*p* = 0.043) and anion gap (*p* = 0.024) (Table 3). Pigs fed Na0 had higher serum phosphorus (*p* < 0.04) and anion gap (*p* < 0.02) than those fed TT, NaT, and KT. However, this increase was still within the reference range for similar aged pigs. Differences between treatment groups were observed in MCH (*p* = 0.024) and segmented neutrophil percentage (*p* = 0.024) (Table 4). Pigs fed NaNa had the highest MCH, greater than that of pigs fed NaT (*p* = 0.003) and KT (*p* = 0.044); those fed TT had higher MCH than those fed NaT (*p* = 0.012). Pigs fed NaT and KT had higher percentages of segmented neutrophils than those fed Na0 and TT (*p* < 0.02), while percentages from those fed NaNa fell intermediary. However, similar to other hematological outcomes, all values still fell within the available reference ranges for both MCH and segmented neutrophils in young pigs.

### 3.4. Tissue Fatty Acid Content

Results from tissue FA analyses are summarized in Table 5, Table 6 and Table 7. Values are presented as percentage of total FA measured.

#### 3.4.1. Cerebral Cortex

Differences were observed in n-6 PUFA 18:2n-6, 20:3n-6, and ARA. Pigs fed NaNa had higher (*p* < 0.001) 18:2n6 than those fed NaT, KT, and TT, followed by pigs fed Na0 who also had higher (*p* < 0.04) 18:2n6 than those fed NaT and TT; pigs fed TT, NaT, and KT all had similar levels. Pigs fed NaNa had higher (*p* ≤ 0.02) 20:3n6 than those fed any other diet. Pigs fed Na0 had higher (*p* ≤ 0.01) ARA than those fed NaT or KT; pigs fed TT, NaT, KT, and NaNa all had similar levels of ARA. The only n-3 present at detectable levels was DHA. Pigs fed TT, NaT, and KT had similar levels of DHA, which were higher (*p* < 0.001) than that of pigs fed Na0 or NaNa; pigs fed NaNa did, however, have higher (*p* < 0.001) levels of DHA than pigs fed Na0. Pigs fed Na0 had higher (*p* < 0.01) total saturated fatty acids (SFA) than those fed any other diet, while pigs fed NaNa only had higher (*p* = 0.023) total SFA than those fed KT. No differences were observed for total monounsaturated fatty acids (MUFA). Overall, there were no differences between TT, NaT, and KT for any FA measured in the cerebral cortex.

#### 3.4.2. Retina

Differences were observed between treatment groups in all n-6 PUFA except 18:3n6. Pigs fed NaT, KT, and TT had similar levels of 18:2n6, all of which were lower (*p* ≤ 0.002) than those of pigs fed Na0 and NaNa; pigs fed Na0 also had lower (*p* < 0.001) 18:2n6 than NaNa. Pigs fed Na0 had higher 20:2n6 than those fed NaT and TT (*p* ≤ 0.05); pigs fed KT and NaNa had higher (*p* ≤ 0.047) 20:2n6 than those fed NaT. Pigs fed NaNa had higher (*p* ≤ 0.014) 20:3n6 than those fed NaT, KT, and TT, but pigs fed Na0 only had higher (*p* = 0.013) 20:3n6 than those fed NaT. Pigs fed Na0 had higher (*p* ≤ 0.026) ARA than those fed NaT, KT, and TT, but levels were similar between TT, NaT, KT, and NaNa. The only n-3 present at detectable levels was DHA. Pigs fed NaT had higher DHA than those fed any other diet (*p* ≤ 0.01), followed by TT and KT which had similar concentrations of DHA. Pigs fed Na0 had lower (*p* ≤ 0.02) concentrations of DHA than those fed NaNa, but both were lower (*p* ≤ 0.01) than that of the other three diets. Pigs fed Na0 and NaNa had higher (*p* < 0.03) total SFA than those fed NaT, KT, and TT; pigs fed KT also had higher (*p* = 0.03) SFA than those fed NaT. Pigs fed Na0 and NaNa had higher (*p* < 0.001) total MUFA than those fed NaT, KT, and TT. Overall, there were no differences between TT and NaT, and the only difference between TT and KT was in DHA.

#### 3.4.3. RBC

ARA was the only n-6 PUFA where differences were observed between treatment groups. Pigs fed NaNa had lower (*p* ≤ 0.002) ARA compared with those fed NaT, Na0, or TT, but similar to pigs fed KT. DHA was the only n-3 PUFA where differences were observed between treatment groups. Pigs fed NaT, KT and TT had similar DHA concentrations, but were higher (*p* < 0.001) than those on diets Na0 and NaNa; pigs fed NaNa also had higher (*p* = 0.006) DHA than those fed Na0. No differences were observed in total SFA, but pigs fed TT had lower (*p* < 0.01) total MUFA than those fed Na0 and NaNa. Overall, there were no differences between TT, NaT, and KT for any FA measured in RBC.

#### 3.4.4. Plasma

ARA was the only n-6 PUFA where differences were observed between treatment groups. Pigs fed NaNa had lower (*p* ≤ 0.015) ARA than those fed any other diet; pigs fed KT also had lower (*p* = 0.048) ARA than those fed Na0. DHA was the only n-3 PUFA where differences were observed between treatment groups. Pigs fed TT and NaT had similar concentrations of DHA, followed by pigs fed KT which had lower (*p* = 0.005) DHA than those fed TT. Pigs on both Na0 and NaNa had lower (*p* < 0.001) DHA concentrations than the other three diets, but those fed NaNa had higher (*p* < 0.001) DHA than those fed Na0. No differences were observed for total SFA, but pigs fed Na0 and NaNa had lower (*p* < 0.04) total MUFA than those fed TT, NaT, and KT. Overall, there were no differences between TT, NaT, and KT in anything other than DHA.

#### 3.4.5. Heart

Differences were observed between treatment groups in all n-6 PUFA except ARA. Pigs fed NaNa had a higher (*p* < 0.04) 18:2n6 than those fed any other diet, and higher (*p* < 0.03) 18:3n6 than those fed NaT, KT, and TT. Pigs fed Na0 also had higher (*p* = 0.02) 18:3n6 than those fed NaT. Pigs on Na0 had lower (*p* ≤ 0.03) 20:2n6 than those fed NaT, KT, and TT, while pigs fed NaNa only had lower (*p* = 0.014) 20:2n6 than those fed TT. Pigs fed TT had higher (*p* < 0.04) 20:3n6 than those fed NaT, Na0, and NaNa; pigs fed KT also had higher (*p* = 0.043) 20:3n6 than those fed Na0. Differences were observed for n-3 PUFA 18:3n3 and DHA. Pigs fed NaNa had higher (*p* < 0.04) 18:3n3 than those fed any other diet. Pigs fed either Na0 or NaNa had similar DHA levels but both were lower (*p* ≤ 0.013) than that of pigs fed NaT, KT, or TT. The latter three had similar DHA levels. No difference were observed for total SFA, but pigs fed Na0 had higher (*p* < 0.05) total MUFA than those fed TT and NaT. Overall, there were no differences between TT, NaT, and KT for any FA except 20:3n6.

#### 3.4.6. Muscle

No differences were observed between treatment groups for any of the n-6 PUFA measured. DHA was the only n-3 PUFA with differences between treatment groups, where pigs fed TT, NaT, and KT all had higher (*p* < 0.001) DHA than those fed Na0 and NaNa. No difference were observed in total SFA, but pigs fed Na0 and NaNa did have higher (*p* < 0.04) total MUFA than those fed NaT, KT, and TT. Overall, there were no differences between TT, NaT, and KT in any FA measured in skeletal muscle.

#### 3.4.7. Liver

No differences were observed between treatment groups for any of the n-6 PUFA measured. Conversely, differences were observed for every n-3 PUFA measured. Pigs fed NaNa had lower (*p* < 0.03) 18:3n3 than those fed NaT, KT, and TT, while pigs fed Na0 only had lower 18:3n3 than those fed KT (*p* = 0.047). Pigs fed Na0 had lower (*p* < 0.001) 20:3n3 than those fed NaT, KT, and TT; the later three and those fed NaNa all had similar levels. Pigs fed NaNa had higher (*p* ≤ 0.022) 20:5n3 than those fed NaT, KT, and TT; the latter three and those fed NaNa all had similar levels. Pigs fed NaT had higher (*p* ≤ 0.030) DHA than those fed Na0 and NaNa, followed by pigs fed KT who only had a higher (*p* = 0.015) DHA than those fed Na0. No difference were observed for total SFA. Pigs fed Na0 did have higher (*p* < 0.04) total MUFA than those fed NaT, KT, or TT, while pigs fed NaNa only had higher (*p* < 0.02) MUFA than those fed TT. Overall, there were no differences between TT, NaT, and KT in any FA measured in the liver.

### 3.5. Liver Histology

No significant differences in liver histology were observed between treatment groups (Appendix A). The only histopathological finding of significance was hepatocellular cytoplasmic vacuolization, but it was noted that the vacuolization observed was likely due to accumulation of glycogen, and not infiltration of lipid. Extra medullary hematopoiesis (EMH) was also observed, but this is not unusual for young animals and would only be alarming if observed in conjuncture with anemia, which was not observed.

## 4. Discussion

Breast milk and IF both provide lipid predominantly in the form of TG. Breast milk does provide some ARA and DHA as PL, but at a much lower concentration relative to the TG form [23]. Exogenous sources of ARA and DHA, both as TG and PL, are available for inclusion in IF in the United States. While FA salts have been used in animal nutrition for some time, specifically in ruminant species [52,53], less is known about the inclusion of fatty acid salts during early human development. The present study evaluated two novel sources of ARA in the form of Na^+^ and K^+^ salts, where we sought to demonstrate safety and efficacy to establish use of Na-ARA and K-ARA in IF. Salts were produced from the parent *M. alpina* oil, an ARA-rich oil used in several GRAS-approved products [28,30,54,55]. Both FA salt forms are intended for use in term IF as part of lipid blends that would supply ARA at a maximum of 0.75% of total dietary lipid content, and are intended to be provided in combination with DHA to produce a dietary DHA:ARA ratio between 1:1 and 1:2. The young pig was used to allow for tissue FA accretion to be quantified. This species is considered an ideal preclinical animal model for the infant due to similarities in tissue growth, brain development, and general metabolism and immunology [36,37,38,39,40,41,42]. Comparisons were made using a control diet that contained both ARA and DHA in TG form, which represent the most frequently supplemented forms in IF. Test articles in diets NaT (Na-ARA) and KT (K-ARA) were provided alongside DHA as TG in accordance with their intended application. A diet including only Na-ARA (Na0) was included as an additional measure to ensure safety even in the absence of DHA. The last diet (NaNa) provided both ARA and DHA in Na-salt form as part of a combined blend containing a 1.5:1 ratio of ARA to DHA.

Multiple parameters were measured to demonstrate the safety of providing these novel FA test articles. We did not observe any concerning changes in safety outcomes, growth performance, or tolerance in the young pig. Additionally, pigs exhibited normal and comparable stool consistency, average daily milk intake, and total body weight gain throughout the study duration. As for tissue growth, only kidney and spleen showed any differences in organ weights, and only when analyzed as relative weights. Compared to control pigs, those fed NaT and NaNa had higher kidney weights, and those fed Na0 and NaNa had lower spleen weights. There does not appear to be a clear trend indicating the inclusion of ARA salts drive these differences, and on an absolute basis, there were no differences in any organ weights between groups. Moreover, all hematological outcomes fell within or just outside identified reference ranges for pigs of similar age [56,57]. Differences in serum phosphorus and anion gap were observed, though these differences appeared to be driven by the lack of DHA inclusion (diet Na0) rather than inclusion of ARA salts, and all values fell within standard reference ranges. Complete blood cell counts showed differences in MCH and percentage of segmented (mature) neutrophils that were driven by treatments NaT and KT. Given that these values were not elevated relative to reference ranges, and that they did not coincide with any other indicators of deteriorating health status, they are likely not biologically significant nor indicative of adverse effects due to ARA salt inclusion. Safety was further supported through histological evaluations on three separate liver lobes, which did not reveal any concerning abnormalities. 

The ARA salts used here dissociate to their mineral and FFA components when mixed with water (e.g., when MR powders are reconstituted). Because FFA in the stomach are shown to inhibit gastric lipase activity, inhibition could be one concern regarding ARA salt supplementation. However, long chain FFA like these only appear to inhibit gastric lipase when generated endogenously from lipolysis [58]. Previous studies with DHA and EPA supplementation as FFA in adults have demonstrated high bioavailability and the potential for a more rapid post-prandial increase in circulating concentrations relative TG forms [34,35]. Given the pro-inflammatory bioactivity of ARA, the potential for a rapid increase in circulating ARA levels that is unbalanced by DHA may be of concern. The present study did not investigate post-prandial circulating ARA kinetics, but at the end of the study, pigs fed either NaT, KT, or Na0 exhibited RBC and plasma ARA concentrations that were similar to those of control pigs (TT). In contrast, pigs fed diet NaNa exhibited reduced circulating ARA concentrations compared to control pigs, which may be more reflective of concentrations within the diet itself, as evident from several tissue FA accretion responses.

Cerebral cortex ARA accretion was responsive to dietary treatment. Similar ARA concentrations were observed among all groups except those fed Na0, which contained no DHA. Pigs on Na0 exhibited numerically higher ARA accretion in the brain compared with pigs consuming DHA-enriched diets, though it should be pointed out that brain ARA concentrations in treatment Na0 were not statistically higher than in control pigs. Similar trends were observed for the retina, where pigs on Na0 exhibited elevated ARA accretion, which in this case was statistically higher than that of control pigs. Stable ARA concentrations in the cerebral cortex and retina across all treatments supplemented with DHA demonstrate equivalent efficacy of Na-ARA and K-ARA as compared to TG-ARA in these key tissues.

Cerebral cortex DHA concentration was also responsive to dietary treatment. Concentrations of DHA for pigs fed NaT or KT were equivalent to control levels, indicating Na-ARA and K-ARA do not differentially influence TG-DHA accretion compared to TG-ARA. Those fed no DHA (Na0) had DHA concentrations that were reduced by 29% compared with the control, which was expected as the brain is sensitive to dietary DHA concentrations and the ARA:DHA ratio [14,15,59]. Those fed the combined Na-ARA and Na-DHA also exhibited an 18% reduction in brain DHA relative to the control, which may reflect differences in dietary intake as levels fell intermediary between Na0 and treatments containing TG-DHA. Similar trends were observed for DHA in the retina. Interestingly, pigs fed NaT exhibited the highest retinal DHA concentration overall. Based on analyzed values, the NaT diet had similar DHA content but higher ARA content than the TT diet. While retinal ARA concentrations between NaT and TT remained stable, DHA concentrations were increased in NaT, suggesting Na-ARA at this level may be responsible for boosting DHA accretion. Overall, the inclusion of Na-ARA and K-ARA did not negatively affect TG-DHA accretion in either cerebral cortex or retina, demonstrating their ability to support accretion in key neural tissues.

Bioequivalence was primarily evaluated in the brain and retina. ARA and DHA are shown to rapidly accumulate in these tissues from the third trimester to two years of age [2] and influence related developmental outcomes including cognitive development and visual acuity [7,9,10,60]. Bioequivalence was based on 90% CI around the geometric LSM and achieved if this interval fell between 80–125% [49,50,51]. It should be noted that ‘bioequivalence’ differs from a statistical difference, as a statistical difference may be present but not clinically relevant. Bioequivalence refers to two products being so similar that they are unlikely to yield clinically relevant differences in therapeutic and/or adverse effects [61]. In both cerebral cortex and retinal tissues, Na-ARA and K-ARA equally supported ARA accretion compared with control TG-ARA levels. Bioequivalence in non-neural tissues was also explored, and outcomes were more variable than neural tissues and produced CI exceeding the limits established for bioequivalence assessment. Differences were most apparent in the heart and liver, and more exaggerated by the absence of dietary DHA. Confidence intervals for secondary tissues were much wider than that observed in cerebral cortex and retina, which could be addressed by increasing replication in future studies.

Because of competition for tissue accretion and generally opposing physiological actions, there is a general consensus in the literature that equal or greater amounts of ARA should be supplemented in conjunction with DHA [8,62,63]. However, there is a dearth of literature to suggest what physiological responses may occur when providing IF containing ARA in the absence of DHA. This is likely a result of the presumed inflammatory risk with increased ARA without a balancing source of DHA. de la Presa Owens previously demonstrated that the inclusion of dietary ARA at 0.8% of FA alone in formula-fed pigs yielded comparable body and organ growth to that of pigs provided an unsupplemented formula, a formula with 0.3% DHA alone, and sow’s milk [64]. The same study used an in vitro assay to demonstrate the provision of ARA alone did not significantly impact hepatic microsomal delta-6 desaturation of 18:2n6 or 18:3n3 compared to an unsupplemented formula or one with DHA alone. ARA alone did result in higher delta-5 desaturation of 18:2n6 and 18:3n-3 compared with unsupplemented pigs [64]. Huang and Craig-Schmidt demonstrated that providing dietary ARA and DHA alone to young pigs resulted in higher and lower ex vivo lung eicosanoid production, respectively, than that of those provided ARA and DHA combined, which fell intermediary [65]. Here, the provision of Na-ARA in the absence of DHA did not prompt any concerning changes in growth or safety parameters, including serum chemistry, complete blood cell count, and liver histology.

Differences were observed between formulated and analyzed dietary ARA and DHA concentrations, though it should be noted that tissue FA outcomes confirmed equivalent accretion rates between control and test diets, thereby supporting our hypothesis. When provided in the TG form, both ARA and DHA were analyzed to contain the correct formulated concentrations, suggesting there were no issues with diet manufacturing. Moreover, all diet samples were collected shortly after initiating the live-animal study and stored in airtight glass containers away from light and at −20 °C until analysis. Given the quality controls put in place to preserve the integrity of the diet samples, we have no definitive evidence to suggest that test articles degraded, and again, tissue incorporation of ARA and DHA support this theory. When comparing the tissue responses of pigs fed NaT, KT, and Na0 to the control, the only tissue in which we observed a difference in ARA accretion was the retina, where those fed Na-ARA without TG-DHA (Na0) had slightly elevated ARA. In comparison, a previous study conducted by our lab observed that a diet purposefully devoid of both ARA and DHA yielded RBC and plasma ARA concentrations of 5.4% and 4.8%, respectively [43]. This is markedly lower than what was observed here in pigs fed diet Na0 (7.1% and 8.7%, respectively), albeit somewhat comparable to that of pigs fed NaNa (5.5% and 5.7%). Thus, based on biological responses observed in our pig study, we conclude that diet analyses did not accurately reflect the concentration of test articles in diets containing salt forms of ARA and DHA, and that they were indeed present near formulated levels at the time of consumption by the pig.

## 5. Conclusions

The inclusion of Na-ARA and K-ARA salts did not elicit differences in growth performance, liver histology, tolerance, daily wellness measures, or hematological outcomes when compared with control pigs receiving TG-ARA. There were few differences in tissue FA accretion observed in pigs fed Na-ARA or K-ARA compared with TG-ARA when provided alongside TG-DHA, demonstrating efficacy of ARA salts when included in combination with TG-DHA. Bioequivalence assessments further supported efficacy, demonstrating Na-ARA and K-ARA equally supported ARA accretion in the cerebral cortex and retina compared with control TG-ARA. Overall, these novel ARA salts appear both safe and efficacious for supporting accretion in critical neural tissues during early development.

## Figures and Tables

**Figure 1 nutrients-13-01482-f001:**
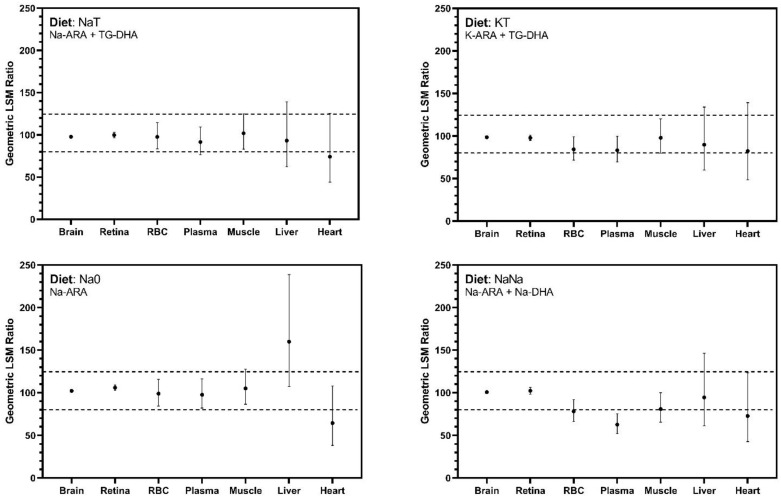
ARA bioequivalence assessments. Geometric least squares mean ratios of tissue ARA from dietary treatment group compared with the control (TT). Values presented as percentages and bioequivalence are met if 90% confidence intervals fall between 80–125% (horizontal dashed lines). All pigs received dietary treatments for 23 days. Abbreviation: ARA, arachidonic acid; DHA docosahexaenoic acid; LSM, least squares mean; TG, triglyceride.

**Table 1 nutrients-13-01482-t001:** Ingredient and nutritional composition of MR powders ^1^.

Treatment	TT	NaT	KT	Na0	NaNa
ARA	TG	Na-Salt	K-Salt	Na-Salt	Na-Salt
Item DHA	TG	TG	TG	-	Na-Salt
Ingredients, % of diet					
Lactose	39.16	39.72	39.69	40.33	40.23
Soy protein isolate ^2^	25.32	25.32	25.32	25.32	25.32
Died coconut oil ^3^	15.17	15.08	15.08	15.13	15.01
Dried MCT oil ^3^	7.59	7.54	7.54	7.57	7.51
Dicalcium phosphate	2.00	2.00	2.00	2.00	2.00
Calcium carbonate	1.98	1.98	1.98	1.98	1.98
Potassium citrate tribasic monohydrate	1.88	1.88	1.88	1.88	1.88
Vitamin and mineral premix ^4^	1.17	1.17	1.17	1.17	1.17
Salt	1.13	1.13	1.13	1.13	1.13
Potassium sorbate	1.00	1.00	1.00	1.00	1.00
Lecithin	0.90	0.90	0.90	0.90	0.90
L-Lysine	0.51	0.51	0.51	0.51	0.51
Choline chloride	0.24	0.24	0.24	0.24	0.24
L-Cystine	0.20	0.20	0.20	0.20	0.20
DL-Methionine	0.15	0.15	0.15	0.15	0.15
Powdered cellulose	0.09	0.09	0.09	0.09	0.09
Palatant ^5^	0.08	0.08	0.08	0.08	0.08
TG-DHA ^6^	0.69	0.69	0.69	-	-
TG-ARA ^6^	0.75	-	-	-	-
Na-ARA ^7^	-	0.32	-	0.32	-
K-ARA ^7^	-	-	0.36	-	-
Na-ARA/Na-DHA mix (1.5:1) ^8^	-	-	-	-	0.60
Nutritional profile ^9^					
Carbohydrates, %	42.4	42.9	42.9	43.5	43.4
Protein, %	24.5	24.5	24.5	24.5	24.4
Fat, %	18.2	18.2	18.2	18.2	18.2
Linoleic acid, %	0.68	0.68	0.68	0.68	0.67
Linolenic acid, %	0.02	0.02	0.02	0.02	0.02
SFA, %	15.76	15.66	15.66	15.72	15.59
MUFA, %	0.72	0.72	0.72	0.72	0.71
PUFA, %	0.14	0.14	0.14	0.14	0.14
Energy, kcal/g	4.31	4.33	4.33	4.36	4.35
Analyzed ^10^, % *w*/*w*					
ARA	0.53 (0.47)	0.76 (0.47)	0.14 (0.47)	0.12 (0.47)	0.01 (0.47)
DHA	0.33 (0.32)	0.32 (0.32)	0.34 (0.32)	ND (0.00)	ND (0.32)

^1^ Diets were manufactured as custom blends formulated by TestDiet (St. Louis, MO, USA). All pigs received allotted treatment from PND 2 to PND 23. Abbreviations: ARA, arachidonic acid; DHA, docosahexaenoic acid; MCT, medium chain triglycerides; MR, milk replacer; MUFA, monounsaturated fatty acids; ND, not detectable; PND, postnatal day; PUFA, polyunsaturated fatty acids; SFA, saturated fatty acids; TG, triglyceride. ^2^ Ardex F, Archer Daniels Midland, Decatur, IL. ^3^ Centennial 72 Coconut IP2 and Vital Blend MCT NG, Sensory Effects, Defiance, OH. ^4^ Custom vitamin and mineral premix provided per gram of complete diet: Ca (12.8 mg), P (7.8 mg), K (10 mg), Mg (1 mg), Na (8.7 mg), Cl (8.5 mg), F (8.1 mcg), Fe (161 mcg), Zn (100 mcg), Mn (46 mcg), Cu (19.2 mcg), Co (0.6 mcg), I (1.18 mcg), Mo (1.02 mcg), Se (0.3 mcg), Vitamin B12 (0.11 mcg), Vitamin K (5 mcg), thiamin (2.7 mcg), riboflavin (13.5 mcg), niacin (60 mcg), pantothenic acid (30 mcg), folic acid (1 mcg), pyridoxine (3 mcg), biotin (0.3 mcg), choline chloride (2.06 mg), ascorbic acid (49.2 mcg), Vitamin A (2.8 IU), Vitamin D3 (6.7 IU), Vitamin E (0.33 IU). ^5^ Luctarom Milky Vanilla, Lucta, Barcelona, Spain. ^6^ Docosahexaenoic acid and arachidonic acid powders; CABIO Bioengineering (Wuhan, China) Co., Ltd. ^7^ Sodium- and potassium-ARA-enriched free fatty acid products; Jost Chemical Co, St. Louis, MO, USA. ^8^ Sodium-ARA/DHA (1.5:1) combined product. ^9^ Based on calculated values from latest ingredient analysis information provided by TestDiet. Nutrients expressed as percent of diet on an as-fed basis. ^10^ Presented as percent of total fatty acids measured by weight. Values are numerical averages from three independent analyses. Samples were analyzed in duplicate or greater within each run for a total of eight individual sample analyses. Target concentrations in parentheses.

**Table 2 nutrients-13-01482-t002:** Growth performance of pigs receiving experimental milk replacers with different sources of ARA ^1.^

Diet	TT	NaT	KT	Na0	NaNa	Pooled	Model
Outcome ARA/DHA ^2^	0.47/0.32	0.47/0.32	0.47/0.32	0.47/0.00	0.47/0.32	SEM	*p*-Value
BW, kg							
Initial	1.79	1.79	1.79	1.75	1.77	0.062	0.969
Final	3.29	3.24	3.31	3.58	3.52	0.244	0.775
PND 3 to 22							
ADG, g/d	71.4	69.5	68.2	84.5	82.2	9.95	0.618
ADMI, g/d	760.8	742.4	786.4	846.9	785.1	64.25	0.751
G:F, g BWG:g milk intake	0.09	0.09	0.08	0.10	0.09	0.009	0.496
Organ weights, g/kg BW							
Whole brain	13.17	13.22	12.86	12.07	12.96	0.833	0.853
Liver	40.43	39.67	38.93	37.62	37.16	1.805	0.441
Longissimus dorsi	6.06	6.75	6.72	7.12	6.69	0.670	0.317
Heart	7.40	7.10	7.41	7.09	7.26	0.257	0.721
Kidneys	7.95 ^b^	8.66 ^a^	7.91 ^b^	7.71 ^b^	8.97 ^a^	0.541	0.003
Spleen	2.02 ^a^	2.24 ^a^	2.03 ^a^	1.93 ^b^	1.73 ^b^	0.111	0.029
Lung	14.97	14.39	14.85	15.33	14.63	0.816	0.314
Retina ^3^	0.07	0.08	0.08	0.06	0.07	0.009	0.510

^a, b^ Means lacking a common superscript letter differ (*p* < 0.05). ^1^ Values represent least square means of 10–16 replicate pigs per treatment. Abbreviations: ADG, average daily body weight gain; ADMI, average daily milk intake; ARA, arachidonic acid; BW, body weight; BWG, body weight gain; DHA, docosahexaenoic acid; G:F, feed efficiency; PND, postnatal day; SEM, standard error of the mean. ^2^ Formulated dietary ARA and DHA content (% of total fatty acids).^3^ Retinal weights were not measured on cohort 1 due to difficulty in whole retina removal, but retinal weights were quantified in cohorts 2–4. Values represent least square means of 7–13 replicate pigs per treatment.

**Table 3 nutrients-13-01482-t003:** Serum chemistry analytes of pigs receiving experimental milk replacers with different sources of ARA ^1^.

Diet	TT	NaT	KT	Na0	NaNa	Pooled	Model	Reference
Outcome ARA/DHA ^2^	0.47/0.32	0.47/0.32	0.47/0.32	0.47/0.00	0.47/0.32	SEM	*p*-Value	Interval ^3^
Creatine, mg/mL	0.49	0.45	0.42	0.51	0.46	0.035	0.118	0.51–1.39
BUN, mg/dL	11.3	10.9	11.2	12.4	12.5	1.02	0.225	4.0–39
Total protein, g/Dl	3.75	3.70	3.77	3.67	3.69	0.105	0.939	2.5–6.6
Albumin, g/dL	1.66	1.54	1.73	1.66	1.74	0.072	0.153	1.9–4.0
Globulin, g/dL	2.07	2.28	2.10	2.00	1.95	0.128	0.301	0.3–1.7^4^
Albumin:globulin ratio	0.84	0.72	0.88	0.86	0.93	0.072	0.211	0.7–2.2
Calcium, mg/dL	9.82	9.70	9.92	10.15	9.91	0.253	0.443	9.9–12.5 ^4^
Phosphorus, mg/dL	9.41 ^b^	8.89 ^b^	9.44 ^b^	10.53 ^a^	9.63 ^ab^	0.627	0.043	6.3–11.5 ^4^
Sodium, mmol/L	140.4	139.5	140.1	141.2	140.9	1.07	0.413	125–147
Potassium, mmol/L	6.83	6.67	7.06	7.01	7.07	0.279	0.737	2.9–4.6
Sodium:potassium ratio	20.8	21.0	20.1	20.6	20.0	0.74	0.816	-
Chloride, mmol/L	107.2	107.2	106.8	107.7	107.2	0.73	0.846	93–108 ^4^
Glucose, mg/dL	99.9	93.7	97.9	107.8	107.3	5.54	0.255	34–159
ALP, U/L	789.5	564.0	719.0	586.7	688.1	87.20	0.100	110–1292
AST, U/L	44.4	32.1	41.3	35.0	29.1	9.23	0.260	13–65
GGT, U/L	38.2	45.9	42.4	37.2	40.1	5.11	0.258	33–94 ^4^
Total bilirubin, mg/dL	0.15	0.13	0.17	0.15	0.14	0.024	0.607	0–0.2 ^4^
CPK, U/L	574.9	402.4	518.4	409.2	358.2	152.0	0.543	153–5427 ^4^
Cholesterol total, mg/dL	120.7	105.4	122.4	116.2	109.5	8.82	0.346	-
GLDH, U/L	2.50	1.32	1.67	1.69	0.86	0.664	0.351	-
Bicarbonate, mmol/L	27.2	26.8	28.1	23.7	26.4	4.56	0.057	19–31 ^4^
Magnesium, mg/dL	2.68	2.55	2.61	2.91	2.70	0.149	0.194	-
Triglycerides, mg/dL	61.6	69.4	62.1	54.8	66.0	7.43	0.620	-
Anion gap ^5^	13.0 ^b^	13.2 ^b^	12.3 ^b^	17.0 ^a^	14.5 ^ab^	4.36	0.024	14–29 ^4^

^a, b^ Means lacking a common superscript letter differ (*p* < 0.05). ^1^ Values represent least square means of 10–13 pigs per treatment group at PND 23. Abbreviations: ALP, alkaline phosphatase; ARA, arachidonic acid; AST, aspartate aminotransferase; BUN, blood urea nitrogen; CBC, complete blood cell count; CPK, creatine phosphokinase; DHA, docosahexaenoic acid; GGT, gamma-glutamyl transferase; GLDH, glutamate dehydrogenase; SEM, standard error of the mean. ^2^ Formulated dietary ARA and DHA content (% of total fatty acids). ^3^ Estimated reference intervals for hematological outcomes for 30-day-old pigs, retrieved from Ventrella et al., 2017, applies to all values unless otherwise indicated. ^4^ Estimated reference intervals for hematological outcomes for 42-day-old pigs, retrieved from Cooper et al., 2014. ^5^ Anion gap reflects difference in serum concentrations of measured cations and anions. Calculated as (Na^+^ + K^+^)-(Cl^−^ + HCO_3_^−^).

**Table 4 nutrients-13-01482-t004:** Complete blood cell counts of pigs receiving experimental milk replacers with different sources of ARA^1^.

Diet	TT	NaT	KT	Na0	NaNa	Pooled	Model	Reference
Outcome ARA/DHA ^2^	0.47/0.32	0.47/0.32	0.47/0.32	0.47/0.00	0.47/0.32	SEM	*p*-Value	Interval ^3^
RBC count, ×10^6^ cells/uL	5.53	5.62	5.55	5.95	5.64	0.337	0.281	4.08–8.17
Hemoglobin, g/dL	9.73	9.36	9.46	10.27	10.04	0.586	0.129	4.32–13.3
Packed cell volume, %	32.31	31.06	31.52	34.17	33.34	1.71	0.120	16–41
MCV, fl	58.63	55.90	57.13	58.89	59.05	1.20	0.119	34.2–61.3
MCH, pg	17.58 ^ab^	16.62 ^c^	17.00 ^bc^	17.28 ^abc^	17.79 ^a^	0.273	0.024	9.4–19.8
MCHC, g/dL	30.09	30.15	29.92	30.05	30.11	0.338	0.921	26.5–33.6
Platelets, ×10^3^ platelets/uL	613	697	643	659	735	36.1	0.145	192–832
WBC count, ×10^3^ cells/uL	15.13	17.39	17.69	15.49	15.27	1.963	0.665	5.6–18.5
Segmented neutrophils, %	40.36 ^b^	50.55 ^a^	50.45 ^a^	40.67 ^b^	44.70 ^ab^	3.014	0.024	10.8–70.6
Band neutrophils, %	0.18	<0.01	<0.01	0.08	0.10	0.084	0.492	-
Lymphocytes, %	52.00	41.55	44.64	51.69	48.50	3.732	0.170	26.2–82.9
Monocytes, %	3.40	6.09	4.09	4.15	5.70	0.777	0.070	1.4–8.3
Eosinophils, %	0.36	0.64	0.36	0.54	0.90	0.223	0.443	0–1.9
Basophils, %	0.19	0.37	0.09	0.06	0.11	0.133	0.241	0–0.90

^a,b,c^ Means lacking a common superscript letter differ (*p* < 0.05). ^1^ Values represent least square means of 10–13 pigs per treatment group at PND 23. Abbreviations: ARA, arachidonic acid; DHA, docosahexaenoic acid; MCH, mean cell hemoglobin; MCHC, mean corpuscular hemoglobin concentration; MCV, mean cell volume; MPV, mean platelet volume; RBC, red blood cell; SEM, standard error of the mean; WBC, white blood cell. ^2^ Formulated dietary ARA and DHA content (% of total fatty acids). ^3^ Estimated reference intervals for hematological outcomes retrieved from Ventrella et al., 2017.

**Table 5 nutrients-13-01482-t005:** Select cerebral cortex and retina FA concentrations of pigs receiving experimental milk replacers with different sources of ARA (% of total measured FA) ^1^.

Diet	TT	NaT	KT	Na0	NaNa	Pooled	Model
FA ARA/DHA ^2^	0.47/0.32	0.47/0.32	0.47/0.32	0.47/0.00	0.47/0.32	SEM	*p*-Value
Cerebral cortex							
∑SFA ^3^	57.61 ^bc^	57.70 ^bc^	57.24 ^c^	59.25^a^	58.07 ^b^	0.282	<0.001
∑MUFA ^4^	15.89	16.34	16.33	16.63	16.86	0.351	0.250
18:2n-6	1.08 ^c^	1.03 ^c^	1.10 ^bc^	1.16 ^ab^	1.22 ^a^	0.032	<0.001
18:3n-6	0.04	0.04	0.04	0.05	0.05	0.004	0.125
20:2n-6	0.13	0.13	0.13	0.14	0.14	0.009	0.675
20:3n-6	0.62 ^b^	0.61 ^b^	0.62 ^b^	0.63 ^b^	0.69 ^a^	0.025	0.009
20:4n-6	14.83 ^ab^	14.51 ^b^	14.63 ^b^	15.17 ^a^	14.95 ^ab^	0.180	0.025
22:2n-6	ND	ND	ND	ND	ND	-	-
18:3n-3	ND	ND	ND	ND	ND	-	-
20:3n-3	ND	ND	ND	ND	ND	-	-
20:5n-3	ND	ND	ND	ND	ND	-	-
22:6n-3	9.87 ^a^	9.69 ^a^	9.56 ^a^	6.99 ^c^	8.06 ^b^	0.330	<0.001
Retina							
∑SFA^3^	52.69 ^bc^	52.31 ^c^	53.30 ^b^	54.96 ^a^	54.33 ^a^	0.365	<0.001
∑MUFA^4^	16.56 ^b^	16.35 ^b^	16.26 ^b^	17.50 ^a^	17.30 ^a^	0.252	<0.001
18:2n-6	1.60 ^c^	1.50 ^c^	1.62 ^c^	1.85 ^b^	2.12 ^a^	0.076	<0.001
18:3n-6	0.12	0.12	0.12	0.14	0.14	0.010	0.063
20:2n-6	0.17 ^bc^	0.16 ^c^	0.18 ^ab^	0.19 ^a^	0.18 ^ab^	0.013	0.047
20:3n-6	0.34 ^bc^	0.31 ^c^	0.34 ^bc^	0.35 ^ab^	0.38 ^a^	0.012	0.005
20:4n-6	11.85 ^b^	11.83 ^b^	11.64 ^b^	12.55 ^a^	12.16 ^ab^	0.264	0.037
22:2n-6	ND	ND	ND	ND	ND	-	-
18:3n-3	ND	ND	ND	ND	ND	-	-
20:3n-3	ND	ND	ND	ND	ND	-	-
20:5n-3	ND	ND	ND	ND	ND	-	-
22:6n-3	16.45 ^b^	17.73 ^a^	16.57 ^b^	12.51 ^d^	13.53 ^c^	0.324	<0.001

^a–c^ Means lacking a common superscript letter differ (*p* < 0.05). ^1^ Values represent least square means of 7–13 replicate pigs per treatment that had received experimental diets over a 21-d period starting at approximately 2 d of age. Fatty acid concentrations were quantified using a validated gas chromatographic technique where C11:0 and C23:0 fatty acids served as internal standards. Abbreviations: ARA, arachidonic acid test article; DHA, docosahexaenoic acid test article; FA, fatty acid; ND, not detected (i.e., below detectable limit); SEM, standard error of the mean. ^2^ Formulated dietary ARA and DHA content (% of total fatty acids). ^3^ ∑SFA = 4:0, 6:0, 8:0, 10:0, 12:0, 13:0, 14:0, 15:0, 16:0, 17:0, 18:0, 20:0, 21:0, 22:0, 24:0. ^4^ ∑MUFA = 14:1n-5, 15:1n-5, 16:1n-7, 17:1n-7, 18:1 trans-n9, 18:1n-9, 20:1n-9, 22:1n-9, 24:1n-9.

**Table 6 nutrients-13-01482-t006:** Select RBC and plasma FA concentrations of pigs receiving experimental milk replacers with different sources of ARA (% of total measured FA) ^1^.

Diet	TT	NaT	KT	Na0	NaNa	Pooled	Model
FA ARA/DHA ^2^	0.47/0.32	0.47/0.32	0.47/0.32	0.47/0.00	0.47/0.32	SEM	*p*-Value
RBC							
∑SFA ^3^	46.60	46.62	46.89	46.47	46.28	0.307	0.578
∑MUFA ^4^	27.92 ^b^	28.80 ^ab^	28.79 ^ab^	30.50 ^a^	30.59 ^a^	0.674	0.018
18:2n-6	14.85	14.05	14.86	14.56	15.17	0.570	0.456
18:3n-6	ND	ND	ND	ND	ND	-	-
20:2n-6	ND	ND	ND	ND	ND	-	-
20:3n-6	0.61	0.53	0.5	0.53	0.51	0.027	0.063
20:4n-6	7.10 ^a^	7.01 ^a^	6.26 ^ab^	7.07 ^a^	5.53 ^b^	0.632	0.004
22:2n-6	ND	ND	ND	ND	ND	-	-
18:3n-3	0.34	0.31	0.38	0.35	0.36	0.029	0.541
20:3n-3	ND	ND	ND	ND	ND	-	-
20:5n-3	ND	ND	ND	ND	ND	-	-
22:6n-3	1.76 ^a^	1.79 ^a^	1.69 ^a^	0.59 ^c^	0.98 ^b^	0.136	<0.001
Plasma							
∑SFA ^3^	49.44	50.97	52.39	50.14	52.54	2.141	0.245
∑MUFA ^4^	16.75 ^b^	16.23 ^b^	15.61 ^b^	18.42 ^a^	18.20 ^a^	0.768	<0.001
18:2n-6	20.7	20.62	20.86	20.85	21.07	0.853	0.992
18:3n-6	ND	ND	ND	ND	ND	-	-
20:2n-6	ND	ND	ND	ND	ND	-	-
20:3n-6	0.65	0.64	0.59	0.65	0.56	0.048	0.487
20:4n-6	8.66 ^ab^	8.03 ^ab^	7.38 ^b^	8.69 ^a^	5.69 ^c^	0.779	<0.001
22:2n-6	ND	ND	ND	ND	ND	-	-
18:3n-3	0.74	0.65	0.75	0.71	0.67	0.050	0.218
20:3n-3	ND	ND	ND	ND	ND	-	-
20:5n-3	ND	ND	ND	ND	ND	-	-
22:6n-3	3.53 ^a^	3.31 ^ab^	2.94 ^b^	0.96 ^d^	1.65 ^c^	0.216	<0.001

^a–d^ Means lacking a common superscript letter differ (*p* < 0.05). ^1^ Values represent least square means of 7–13 replicate pigs per treatment that had received experimental diets over a 21-d period starting at approximately 2 d of age. Fatty acid concentrations were quantified using a validated gas chromatographic technique where C11:0 and C23:0 fatty acids served as internal standards. Abbreviations: ARA, arachidonic acid test article; DHA, docosahexaenoic acid test article; FA, fatty acid; ND, not detected (i.e., below detectable limit); SEM, standard error of the mean. ^2^ Formulated dietary ARA and DHA content (% of total fatty acids). ^3^ ∑SFA = 4:0, 6:0, 8:0, 10:0, 12:0, 13:0, 14:0, 15:0, 16:0, 17:0, 18:0, 20:0, 21:0, 22:0, 24:0 ^4^ ∑MUFA = 14:1n-5, 15:1n-5, 16:1n-7, 17:1n-7, 18:1 trans-n9, 18:1n-9, 20:1n-9, 22:1n-9, 24:1n-9.

**Table 7 nutrients-13-01482-t007:** Select heart, muscle, and liver FA concentrations of pigs receiving experimental milk replacers with different sources of ARA (% of total measured FA) ^1^.

Diet	TT	NaT	KT	Na0	NaNa	Pooled	Model
FA ARA/DHA ^2^	0.47/0.32	0.47/0.32	0.47/0.32	0.47/0.00	0.47/0.32	SEM	*p*-Value
Heart							
∑SFA ^3^	48.70	51.06	48.26	51.18	48.07	4.523	0.123
∑MUFA ^4^	16.52 ^b^	17.11 ^b^	17.26 ^ab^	18.41 ^a^	17.47 ^ab^	1.390	0.049
18:2n-6	20.79 ^b^	20.25 ^b^	20.31 ^b^	19.69 ^b^	23.13 ^a^	2.685	0.023
18:3n-6	0.102 ^bc^	0.088 ^c^	0.103 ^bc^	0.128 ^ab^	0.134 ^a^	0.015	0.018
20:2n-6	0.45 ^a^	0.42 ^ab^	0.42 ^ab^	0.34 ^c^	0.36 ^bc^	0.041	0.012
20:3n-6	0.82 ^a^	0.64 ^bc^	0.76 ^ab^	0.62 ^c^	0.66 ^bc^	0.111	0.028
20:4n-6	11.01	9.14	11.24	9.11	9.13	2.612	0.179
22:2n-6	ND	ND	ND	ND	ND	-	-
18:3n-3	0.25 ^b^	0.23 ^b^	0.23 ^b^	0.22 ^b^	0.28 ^a^	0.077	0.006
20:3n-3	ND	ND	ND	ND	ND	-	-
20:5n-3	0.25	0.29	0.25	0.21	0.24	0.052	0.628
22:6n-3	1.19 ^a^	1.01 ^a^	1.41 ^a^	0.07 ^b^	0.37 ^b^	0.308	<0.001
Muscle							
∑SFA ^3^	62.12	60.19	60.50	59.23	60.97	1.390	0.623
∑MUFA ^4^	16.87 ^b^	17.41 ^b^	17.22 ^b^	19.14 ^a^	19.27 ^a^	0.652	0.010
18:2n-6	13.39	14.37	13.66	14.21	13.39	0.935	0.886
18:3n-6	0.11	0.11	0.11	0.15	0.12	0.012	0.053
20:2n-6	0.29	0.32	0.3	0.26	0.24	0.030	0.156
20:3n-6	0.6	0.65	0.63	0.63	0.59	0.056	0.931
20:4n-6	5.33	5.7	5.67	5.79	4.87	0.560	0.753
22:2n-6	ND	ND	ND	ND	ND	-	-
18:3n-3	0.36	0.35	0.35	0.37	0.35	0.023	0.840
20:3n-3	ND	ND	ND	ND	ND	-	-
20:5n-3	0.11	0.11	0.11	0.11	0.12	0.014	0.965
22:6n-3	0.88 ^a^	0.88 ^a^	0.88 ^a^	0.16 ^b^	0.30 ^b^	0.066	<0.001
Liver							
∑SFA ^3^	78.55	75.72	75.77	74.02	73.81	2.165	0.186
∑MUFA ^4^	13.59 ^c^	14.23 ^bc^	14.86 ^bc^	16.64 ^a^	15.78 ^ab^	0.772	0.006
18:2n-6	3.93	4.07	4.34	4.36	3.72	0.458	0.784
18:3n-6	0.08	0.09	0.09	0.13	0.11	0.014	0.100
20:2n-6	0.16	0.16	0.17	0.13	0.12	0.018	0.208
20:3n-6	0.18	0.21	0.21	0.22	0.22	0.029	0.764
20:4n-6	2.03	2.39	2.47	3.73	2.7	0.538	0.164
22:2n-6	0.01	0.01	0.01	0.01	0.01	0.003	0.780
18:3n-3	0.10 ^ab^	0.11 ^a^	0.11 ^a^	0.07 ^bc^	0.06 ^c^	0.014	0.029
20:3n-3	0.03 ^a^	0.03 ^a^	0.03 ^a^	0.01 ^b^	0.02 ^ab^	0.004	<0.001
20:5n-3	0.05 ^b^	0.05 ^b^	0.05 ^b^	0.08 ^ab^	0.11 ^a^	0.017	0.038
22:6n-3	0.80 ^abc^	1.01 ^a^	0.96 ^ab^	0.42 ^c^	0.49 ^bc^	0.171	0.026

^a,b,c^ Means lacking a common superscript letter differ (*p* < 0.05). ^1^ Values represent least square means of 5–13 replicate pigs per treatment that had received experimental diets over a 21-d period starting at approximately 2 d of age. Fatty acid concentrations were quantified using a validated gas chromatographic technique where C11:0 and C23:0 fatty acids served as internal standards. Abbreviations: ARA, arachidonic acid test article; DHA, docosahexaenoic acid test article; FA, fatty acid; ND, not detected (i.e., below detectable limit); SEM, standard error of the mean. ^2^ Formulated dietary ARA and DHA content (% of total fatty acids). ^3^ ∑SFA = 4:0, 6:0, 8:0, 10:0, 12:0, 13:0, 14:0, 15:0, 16:0, 17:0, 18:0, 20:0, 21:0, 22:0, 24:0 ^4^ ∑MUFA = 14:1n-5, 15:1n-5, 16:1n-7, 17:1n-7, 18:1 trans-n9, 18:1n-9, 20:1n-9, 22:1n-9, 24:1n-9.

## Data Availability

The data that support the findings of this study are available from the corresponding author upon reasonable request.

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
