# Peer review of "Safety and Efficacy of Sodium and Potassium Arachidonic Acid Salts in the Young Pig"

_nutrients, 2021, doi:10.3390/nu13051482_

Round 1
Reviewer 1 Report
In this manuscript, the authors replaced ARA (and DHA) in triglyceride forms by Na and K salts in milk replacement formulations fed to piglets and then compared health outcomes resulting form these substitutions. The introduction well written and well referenced. M&M is well described and appropriate. However, a major issue will need to be properly addressed before this manuscript is suitable for publication.
Major issue
In Table 1, measured ARA and DHA levels deviated significantly from target levels, especially for NaNa. Also, it is not clear what is the difference between the last two rows of Table 1 and Table S1 (which is not introduced anywhere in the text). Explanations from the last paragraph (L. 570-589, which should also have been provided much earlier in the manuscript) are not very convincing, as I am assuming that all MR formulations were collected and stored in the same condition until analysis. Hence, it is unclear why the ARA and DHA would have degraded to a much larger extent in the NaNa than the other salt treatment groups. Contrary to the authors claim, the comparison with an older study using a diet devoid of ARA and DHA [65] (L. 582-586) does seem to suggest that ARA and DHA levels were indeed much lower than the target concentrations in the NaNa treatment group. The discussion on this issue is deficient and must be strengthened.
Minor comments:
Ideally the abstract should be easily understandable without referring to the manuscript. L. 22: Please specify 8 males and 8 females per treatment group. L.23-25: This sentence is difficult to read and contains a lot of abbreviation including TT which is not yet defined. Please rewrite to clarify. L. 30: “100% bioequivalent” may be a bit too strong. It is probably more appropriate to claim that the effects of the salt forms were undistinguishable from those of the triglyceride forms in the cerebral cortex and retina, given that other tissues presented wider CI ranges that exceeded the 80-125% threshold.
L. 114-115: To be clear piglets received a second antibiotic shot on PND 10 or received only one antibiotic shot at PND 10? Any noticeable difference with this cohort?
L. 178-179: Please correctly format the Lepage and Roy, 1986 reference.
Table 1: The authors may also want to specify the origin of Na-ARA and K-ARA in the main text in addition to Table 1 legend. Please properly format the two “ND” cells on the last DHA row of the table.
L. 264: Please delete the extra dot.
L. 563: Please move reference [63] to line 559.
L. 708: I could not find this reference, please double-check.
Author Response
Please see the attachment where we have responded to each point raised by the reviewer as denoted in red-colored text.

Reviewer 2 Report
The authors presented a well written manuscript that described an investigation into the use the salt forms of ARA and DHA as supplements in a young pig model. The authors presented data that the salts were nutritionally bioequivalent to TG form based on the end points discussed. The authors did not highlight any evidence of lipid profile alteration due to the form of the ARA and DHA use in the feed. The data was nicely presented; however, the manuscript can be improved with the following suggestions:
Although FFAs do not require hydrolysis, the salt tend to be less stable. Although precautions were taken in the experimental design to minimize degradation, little background or discussion was provided concerning the compounds stability. The manuscript could be strengthened by including additional benefits for use of salt forms over the TG forms for ARA and DHA supplementation.
For tables 5-7, clarify the meaning of the superscript (a-d combinations) as well as the model p value (what was the comparison) in the tables.
The bioequivalence threshold established by the authors was not observed for the secondary tissues, including RBC, plasma heart for Na-ARA and Na-DHA. The spread in the data was observed greater for tissues other than brain and retina. The author should comment on potential technical variance that would help understand the cause of the increased variation for those tissues.
Did the authors observe any sex related trends in the study?
Author Response

(The authors gave the same response as above.)

Round 2
Reviewer 1 Report
The authors satisfactorily addressed most of the minor issues. The reason behind the low levels of ARA measured in the NaNa diet is still unclear and the explanation unconvincing. However, the NaNa diet was not the main point of this investigation and hence, this manuscript will be suitable for publication.